# Bariatric Surgery Induced Changes in Blood Cholesterol Are Modulated by Vitamin D Status

**DOI:** 10.3390/nu14102000

**Published:** 2022-05-10

**Authors:** Joanna Reczkowicz, Adriana Mika, Jędrzej Antosiewicz, Jakub Kortas, Monika Proczko-Stepaniak, Tomasz Śledziński, Konrad Kowalski, Łukasz Kaska

**Affiliations:** 1Department of Bioenergetics and Physiology of Exercise, Medical University of Gdansk, 80-210 Gdansk, Poland; joanna.reczkowicz@gumed.edu.pl; 2Department of Pharmaceutical Biochemistry, Medical University of Gdansk, 80-211 Gdansk, Poland; adriana.mika@gumed.edu.pl (A.M.); tomasz.sledzinski@gumed.edu.pl (T.Ś.); 3Department of Health and Natural Sciences, Gdansk University of Physical Education and Sport, 80-336 Gdansk, Poland; jakub.kortas@awf.gda.pl; 4Department of General, Endocrine and Transplant Surgery, Faculty of Medicine, Medical University of Gdansk, Smoluchowskiego 17, 80-214 Gdansk, Poland; monika.proczko-stepaniak@gumed.edu.pl; 5Masdiag Sp. z o.o. Company, Stefana Żeromskiego 33, 01-882 Warsaw, Poland; konrad.kowalski@masdiag.pl

**Keywords:** OAGB, serum lipids, 24,25(OH)_2_D_3_, 25OHD_3_, 25OHD_2_

## Abstract

The effect of metabolically active bariatric surgery treatment on lipid metabolism is inconclusive. The authors of this study presume that initial vitamin D status may play a regulating role in influencing the beneficial post-effects of bariatric surgery, especially the lipid profile. The biochemical data obtained from 24 patients who had undergone laparoscopic one-anastomosis gastric bypass (OAGB) at baseline, 3 months before the surgery, at the time of surgery, and 6 months later, demonstrate that vitamin D status influenced the postoperative lipid profile. The baseline established the partition line which divided patients into two groups according to the stated calcidiol initial concentration level of 32 ng/mL. The data shows that OAGB induces a decrease in TG and hsCRP while increasing HDL. Conversely, in patients whose 25(OH)D_3_ was below 32 ng/mL TC significantly increased while those above this concentration remained in the normal physiological range. The changes induced by OAGB in TG, glucose, and hsCRP were similar in both groups. Unexpectedly, the surgery did not affect vitamin D metabolites. In conclusion, the results of the study suggest that a higher concentration of serum 25(OH)D_3_ may enhance the protective effects of OAGB.

## 1. Introduction

Bariatric surgery is an effective treatment for morbid obesity and can lead to improvement in several cardiovascular risk factors such as hypertriglyceridemia, hypertension, diabetes, oxidative stress, and others [1]. There are several reports demonstrating improvement in all parameters of the lipid profile after bariatric surgery [2,3,4], conversely, some other reports showed no change, or changes only in TG [5,6]. The discrepancy between studies is unknown, possibly factors such as diet, exercise, and others play a relevant role. In this study, we hypothesized that initial vitamin D status may play a role and have a modifying effect on post-bariatric surgery-induced changes in lipid profile.

There are some reports demonstrating that Vitamin D supplementation can modify lipid profiles in young and elderly subjects who had undergone exercise training [7,8]. Conversely, vitamin D alone at a dose of 400 IU/day for 5 years did not affect blood lipids [9]. Vitamin D is fat-soluble, and it is known to accumulate in adipose tissue. Thus, an increase in fat mass leads to a drop-in serum vitamin D which can limit its availability for other tissues [10]. It is generally accepted that obese persons need a double dose of vitamin D supplements to keep their blood levels normal. Modifying the effects of vitamin D on blood lipids will certainly be dose-dependent, unfortunately in some supplementation studies its serum level has not been measured [9] thus the dose amount has not been established.

Apolipoprotein A-I (apo A-I) is a component of high-density lipoprotein (HDL) molecules, and its higher concentration is associated with a lower risk of myocardial infarction [11]. It has been shown that A-I (apo A-I) gene expression is under the control of 1,25-dihydroxy vitamin D_3_ [1,25(OH)_2_D_3_] which is an active form of vitamin D_3_. Consistently, plasma apo A-I is positively correlated with 25(OH)D_3_ levels in both men and women [12].

Vitamin D—produced in the skin from sun exposure or ingested in the diet, has to be activated by two-step reactions catalyzed by vitamin D-25-hydroxylase(s) (CYP2R1, CYP27A1) to 25(OH)D_3_. 25(OH)D_3_. It is then converted by 25-hydroxy-vitamin D-1 alpha-hydroxylase (CYP27B1) to a biologically active form. Moreover, 25(OH)D_3_ and 1,25(OH)_2_D_3_ can undergo hydroxylation at carbon 24 by D-24-hydroxylase (CYP24A1) which leads to their inactivation and formation of 24,25(OH)_2_D_3_ and 1,24,25(OH)_3_D_3_, respectively. Interestingly, 24,25(OH)_2_D_3_ and 3-epi-25(OH)D_3_ which are considered to be inactive forms of vitamin D do have some activity. For example, 24,25(OH)_2_D_3_ protects cells from 1,25(OH)_2_D_3_ toxicity and augments antioxidant potential [13]. A higher concentration of 3-epi-25(OH)D_3_ was associated with an improved cardiovascular risk profile [14]. As mentioned above, bariatric surgery can also lead to a decrease in markers of oxidative stress and upregulate reduced glutathione (GSH) levels in the blood [1]. Conversely, low GSH levels and oxidative stress can modify vitamin D metabolism. Thus, one of the goals of this study was to evaluate the effects of bariatric surgery on vitamin D metabolites as well as on other metabolic parameters.

The worldwide adopted method of bariatric treatment, gastric bypass, especially its modification, one-anastomosis gastric bypass (OAGB) has been approved as one of the strongest metabolically active procedures [15,16]. The main bariatric and metabolic mechanisms of OAGB result from the mixed restrictive and malabsorptive types of anatomical alterations. While the restriction of the working stomach volume to a narrow, long, lesser curvature—pouch provides food intake reduction and depresses ghrelin activity, excluding the long jejunal limb from alimentary transit leads to the enhancement of the incretin reaction and increases portal bile acid reabsorption. Gastrointestinal hormone activity alterations influence the improvement of carbohydrate and lipid metabolism [17]. Metabolic changes induced by OAGB are not homogenous, thus, there must be some factor that can modify this response. In this study, it is observed that the vitamin D status significantly modified changes in serum lipids induced by the OAGB.

## 2. Methods

### 2.1. Patients

The study included 24 patients (4 male, 20 female; age: 43.29 ± 8.74 yrs; weight: before the surgery 117.33 ± 17.46 changed to 107.24 ± 16.74 (kg) after the surgery; BMI: before the surgery 41.44 ± 3.32 changed to 37.85 ± 3.57 (kg/m^2^) after the surgery) with morbid obesity that underwent surgical treatment with OAGB at the Department of General, Endocrine and Transplant Surgery, at the Medical University of Gdansk, between the years 2016 and 2018. The patients on lipid-lowering drugs were excluded from the study. Routine laboratory parameters were determined at the Central Clinical Laboratory, at the Medical University of Gdansk. Blood samples were drawn after an overnight fast and serum was obtained after centrifugation and stored at −80 °C until analysis. Anthropometric and laboratory parameters were measured 3 months before surgery (reference point A), immediately before surgery (reference point B), and again 6 months after the surgery (reference point C).

Patients were qualified and prepared for the OAGB following IFSO criteria for morbidly obese individuals [18]. An adequate preoperative preparation program has been provided by the multidisciplinary team. All of the operations have been performed laparoscopically. The sleeve-shaped 25 cm long pouch was created based on the 36F calibration tube. The average volume of the reservoir has been calculated at 80 mL. The length of the bypassed intestine loop was 180 cm measured from the Treitz ligament to the pouch-jejunal anastomosis. The linear 2.5 cm anastomosis was performed, connecting the narrow pouch with the jejunum.

Patients after OAGB have been supplemented equally with a set of vitamins (“Fit for me WLS Forte” FitForMe B.V. Holland) including 75 µg of D_3_ and followed a special postoperative diet based mainly on proteins. The multidisciplinary team met all the patients in the established follow-up module.

The study was conducted in compliance with the Declaration of Helsinki of the World Medical Association and the protocol was approved by the Local Bioethics Committee at the Medical University of Gdansk (approval no. NKBBN/493/2016). Written, informed consent was obtained from all participants before the study.

### 2.2. Biochemical Analyses

The blood samples were collected for the analysis after an overnight fast at baseline 3 months before surgery (A), at the time of surgery (B), and 6 months post-surgery (C). Routine laboratory parameters were determined at the Central Clinical Laboratory, at the Medical University of Gdansk. The sampling at reference points B and C included lipid profile, hsCRP, glucose, albumin, and protein. To determine all forms of Vitamin D such as 25(OH)D3, 25(OH)D2, 24,25(OH)2D3, and epi-25(OH)D3 the blood samples were prepared for these specific measurements through precipitation and derivatization [19,20]. LC-MS analysis was performed using ExionLC analytical HPLC system with CTC PAL autosampler (Zwinger, Switzerland) coupled with a QTRAP^®^ 4500 MS/MS system (Sciex, Framingham, MA, USA). All data were acquired according to the determined protocol [19].

### 2.3. Statistical Analysis

Statistical analyses were performed by using a statistics software package (Statistica 13.1 software, TIBCO Software, Palo Alto, CA, USA). Shapiro–Wilk tests were used to assess the homogeneity of dispersion from a normal distribution. The Brown–Forsythe test was used to evaluate the homogeneity of variance. One–way analysis of variance was used to evaluate the changes in three measurement points. In a further step of the analysis, to check how the differences in the process impacted on the groups with low and high levels of 25(OH)D_3_ repeated measures analyses of variances (rANOVA) were calculated. In case of a significant time x group interaction, post hoc tests for unequal sample sizes were performed to identify significantly different results. The 95% confidence interval for changes within each of the studied groups was calculated. Additionally, analyses of covariance (ANCOVA) were computed to adjust between-measurements effects for potential differences in weight and BMI [21]. To estimate the practical relevance of the ANCOVA effect sizes (partial eta squared, $\eta_{p}^{2}$) were additionally calculated. According to Cohen et al. [22], an $\eta_{p}^{2}\;\geq 0.001$ indicates a small, ≥0.06 a medium, and ≥0.14 a large effect. Spearman correlation was also performed in order to check the existence of a relationship between the biochemical parameters, weight, and BMI of the study participants. The level of significance was set at *p* < 0.05.

## 3. Results

### Biochemical Data

To establish the partition line of calcidiol at the time point of 3 months before the surgery (reference point A) the data of all forms of Vitamin D such as 25(OH)D_3_, 25(OH)D_2_, 24,25(OH)_2_D_3_ and epi-25(OH)D_3_ were analyzed, resulting in the concentration level of 25(OH)D_3_ at 32 ng/mL (Table 1). There is an obvious correspondence with earlier studies which is clarified in the discussion.

Samples obtained at reference points B and C present the changes in lipid profile and metabolic parameters (Table 2).

It is expressly observed that the concentration of the inflammation marker hsCRP, decreased significantly 6 months after the surgery. After the surgery, LDL and total cholesterol did not change and at the same time, HDL cholesterol increased. Conversely, TG decreased by approximately 40% in a 6 month period of follow-up (delta change: −42.73). Similar results were found in the levels of glucose, with a significant decline (delta change: −29.01). Concentrations of albumin and protein significantly increased. Bariatric surgery had no effects on measured forms on vitamin D_3_, conversely, a decrease in 25(OH)D_2_ was observed.

In the ANCOVA analysis, taking baseline weight values into account, we found small interaction effects on LDL (*p* = 0.80; $\eta_{p}^{2}$ = 0.01) and medium effects on CHOL (*p* = 0.07; $\eta_{p}^{2}$ = 0.08). Taking BMI values into account, pairwise comparison of LDL revealed small interaction effects on LDL (*p* = 0.81; $\eta_{p}^{2}$ = 0.01) and medium effects on CHOL (*p* = 0.10; $\eta_{p}^{2}$ = 0.07). The analysis of the remaining parameters did not show any differences in terms of weight and BMI, the trend of changes remained unchanged. There were no significant correlations between the changes in biochemical parameters, the weight and BMI of participants (Appendix A).

To evaluate the effect of calcidiol on metabolic parameters, according to the assumed concentration level at 32 ng/mL, patients were divided into two groups over 32 ng/mL (*n* = 15) and under 32 ng/mL (*n* = 9), hence the following data were analyzed.

The initial level of 25(OH)D_3_ (calcidiol) assessed at baseline over 32 ng/mL affected the biochemistry data changes induced by the surgery, obtained in the 6 months following surgery.

A benefit of calcidiol levels over 32 ng/mL explicitly affected cardiometabolic risk scorekeeping cholesterol levels in the normal range (delta change: −4.65, *p* > 0.05) while in group <32 ng/mL it was associated with a significant increase (delta change: 43.61, *p* < 0.02). An analogical effect was observed in the level of LDL in the >32 ng/mL group; it was finally reduced (delta change: −14.68, *p* > 0.05) while in the <32 ng/mL group it increased (delta change: 12.92, *p* > 0.05). No other significant associations were found. It must be further mentioned that the calcidiol >32 ng/mL group is associated with the levels of hsCRP, HDLC, and TG being lower immediately before surgery, concerning the analogical levels of patients with initial calcidiol in the group under 32 ng/mL, although the tendency remained the same (Table 3).

No significant correlations between baseline values of Vitamin D metabolites and metabolic parameters were noticed (Appendix A).

## 4. Discussion

The current study defined whether bariatric surgery-induced changes in lipid profile are vitamin D dependent and whether vitamin D influences the metabolism. First of all, we observed several positive changes in the blood biochemistry of our patients. Fasting glucose and TG concentration decreased and serum protein increased. Besides, we demonstrated that serum 25(OH)D_3_, 24,25(OH)_2_D_3_, and 3-epi-25(OH)D_3_ did not change six months after the surgery. According to several reports, bariatric surgery has positive effects on serum lipids as the decrease in blood cholesterol, LDL, and TG with a concomitant increase in HDL has been observed [2,3,4]. Conversely, some studies demonstrated no changes in serum lipids after the surgery [5]. All of these data indicate that there must be some modifying factor.

Similarly, we observed an increase in HDL and a decrease in TG but no significant changes in TC and LDL after 6 months of follow-up. These data let us ponder if vitamin D status could play a role in bariatric surgery-induced changes in lipid profile. Vitamin D deficiency is a worldwide problem that impacts around one billion people. Deficiency or insufficiency of vitamin D is mainly caused by inadequate dietary intake, sedentary lifestyles, and reduced sun exposure [23]. It is important to note that obesity is one of the factors which predispose to vitamin D deficiency, where levels < 30–35 ng/mL are viewed as insufficient [24]. Some researchers suggested that serum concentration of 25(OH)D_3_ range 50–75 ng/mL is an optimal level [14].

Previously, a relevant drop in TC has been demonstrated in training young rowing athletes, who were accompanied by vitamin D supplementation, which increased blood 25(OH)D_3_ above 32 ng/mL. At the same time, training without supplementation had no effects on blood lipids [7]. A similar corresponding observation has been noted on senior subjects training in Nordic Walking [8]. Surprisingly among our patients, only nine of them had a serum concentration level of 25(OH)D_3_ above 32 ng/mL. When they were divided into two groups based on baseline serum concentration of 25(OH)D_3_, distinct effects of the surgery on lipid profiles were observed. We have found a significant increase in TC and a trend to increase in LDL and HDL (not significant) in patients whose 25(OH)D_3_ was below 32 ng/mL. Conversely, in patients with a serum concentration level of 25(OH)D_3_ above 32 ng/mL, no change in TC (which was in the normal range) and a tendency to decrease in LDL was observed. However, a decrease in serum glucose, TG, and an increase in plasma proteins were similar in both groups. These data are coherent with several reports demonstrating that different levels of concentration of serum 25(OH)D_3_ are needed to obtain its protective effects. For example, to protect children from rickets only 20 ng/mL is sufficient [25] while to reduce the risk of type I diabetes more than 40 ng/mL is necessary [26]. There are several studies where the effects of vitamin D on lipid profile have been investigated. However, most of these studies showed no effect of vitamin D supplementation on lipid profile. For example, in New Zealand, insulin-resistant patients who received 4000 IU of vitamin D or placebo had no observed effect of vitamin D treatment on lipid profiles [27]. Similarly, in another study, supplementation of vitamin D 5000 IU for 12 weeks did not induce changes in HDL and LDL [28]. Conversely, there are some studies where the effects of vitamin D supplementation on blood concentration of 25(OH)D_3_ were minute, thus making it difficult to draw conclusions [29].

As mentioned above both oxidative stress and low levels of GSH, which often occur in obese individuals, can inhibit the activity of vitamin D hydroxylase responsible for its activation, while it can also upregulate CYP24A1, an enzyme inactivating vitamin D [13]. Thus, we hypothesized that bariatric surgery may influence vitamin D metabolism by increasing the formation of inactive forms of vitamin D. However, this was not the case, the only difference was a lower concentration of 25(OH)D_2_ in the 6 months period after surgery. 25(OH)D_2_ is of dietary origin, therefore it may be the result of a diet change or impaired intestinal absorption.

## 5. Conclusions

In conclusion, this is the first report demonstrating that bariatric surgery-induced changes in serum lipid profile are significantly modulated by vitamin D status. We believe that our data can help to explain why some studies showed negative effects of bariatric surgery on blood cholesterol while others showed the contrary. Moreover, the data indicate that supplementation of vitamin D which is often required among bariatric patients should be adequate to increase the concentration of 25(OH)D_3_ above the partition level of 32 ng/mL before OAGB at least in the 3 months before surgery time point.

## Figures and Tables

**Table 1 nutrients-14-02000-t001:** Vitamin D metabolites results were measured at all sampling points.

	A	B	C	One-Way ANOVA
25(OH)D3 [ng/mL]	27.55 ± 10.89	29.88 ± 13.08	33.97 ± 11.43	0.08
25(OH)D2 [ng/mL]	0.45 ± 0.32	0.31 ± 0.23	0.14 ± 0.12 *	0.00
24,25(OH)2D3 [ng/mL]	2.13 ± 1.32	2.48 ± 1.62	2.51 ± 1.41	0.21
epi-25(OH)D3 [ng/mL]	1.14 ± 0.78	1.31 ± 0.91	1.57 ± 0.98	0.11

Note: one-way ANOVA, variance comparing significant differences. * Statistically significant difference from A and B.

**Table 2 nutrients-14-02000-t002:** Biochemistry—analyzed data reference points: B vs. C.

	B	C	Δ (CI)	*p*
hsCRP [mg/L]	2.78 ± 1.24	1.73 ± 0.58	−1.05 (−1.56;−0.55)	0.00
LDL [mg/dL]	116.33 ± 31.24	118.9 ± 29.31	2.57 (−15.24;20.38)	0.77
HDLC [mg/dL]	41.04 ± 8.14	50.01 ± 11.63	8.98 (4.14;13.81)	0.00
CHOL [mg/dL]	172.69 ± 35.56	198.2 ± 38.33	25.51 (3.46;47.56)	0.09
TG [mg/dL]	158.09 ± 75.71	115.36 ± 44.84	−42.73 (−73.54;−11.92)	0.01
GLU [mg/dL]	131.4 ± 20.79	102.39 ± 20.37	−29.01 (−41.52;−16.5)	0.00
ALB [g/L]	37.17 ± 2.26	41.45 ± 3.89	4.28 (2.57;5.99)	0.00
PRO [g/L]	61.29 ± 5.89	66.58 ± 5.53	5.29 (2.18;8.4)	0.00

Note: hsCRP, C-reactive protein highly sensitive; LDLC, low-density lipoprotein cholesterol; HDLC, high-density lipoprotein cholesterol; CHOL, cholesterol; TG, triglycerides; GLU, glucose; ALB, albumin; PRO, protein; Δ (CI), confidence intervals change; *p*, statistically significant difference.

**Table 3 nutrients-14-02000-t003:** Effects of bariatric surgery on analyzed parameters—the role of vitamin D. Analysis of Biochemistry data between reference points B vs. C divided into two groups considering the baseline level of 25(OH)D3 at 32 ng/mL.

	(A) 25(OH)D3 < 32 [ng/mL] (*n* = 15)			(A) 25(OH)D3 > 32 [ng/mL] (*n* = 9)			rANOVA
	B	C	Δ (CI)	B	C	Δ (CI)	Group *×* Time
hsCRP mg/L]	3.06 ± 1.27	1.85 ± 0.63	−1.21 (−1.97;−0.45)	2.31 ± 1.1	1.52 ± 0.45	−0.8 (−1.44;−0.15)	0.42
LDL [mg/dL]	115.8 ± 34.9	128.72 ± 31.78	12.92 (−12.96;38.79)	117.21 ± 25.92	102.54 ± 14.84	−14.68 (−35.84;6.49)	0.12
HDLC [mg/dL]	43.69 ± 8.05	53.34 ± 13.23	9.66 (2.48;16.83)	36.62 ± 6.47	44.46 ± 5.25	7.84 (0.95;14.74)	0.72
CHOL [mg/dL]	169.9 ± 35.85	213.51 ± 39.6 *	43.61 (13.78;73.45)	177.34 ± 36.71	172.69 ± 17.54	−4.65 (−30.41;21.1)	0.02
TG [mg/dL]	160.7 ± 86.49	124.44 ± 52.76	−36.26 (−82.96;10.45)	153.73 ± 57.93	100.21 ± 22.29	−53.51 (−93.08;−13.94)	0.59
GLU [mg/dL]	126.81 ± 21.14	103.83 ± 22.71	−22.99 (−38.85;−7.12)	139.04 ± 18.85	99.99 ± 16.73	−39.04 (−62.19;−15.9)	0.21
ALB [g/L]	37.11 ± 2.46	41.97 ± 4.56	4.86 (2.41;7.3)	37.27 ± 2.03	40.58 ± 2.4	3.31 (0.72;5.89)	0.37
PRO [g/L]	61.58 ± 7.26	67.77 ± 5.35	6.19 (1.6;10.78)	60.79 ± 2.66	64.58 ± 5.54	3.79 (−0.52;8.1)	0.45

Note: CRPHS, C-reactive protein highly sensitive; LDL, low-density lipoprotein; HDLC, high-density lipoprotein cholesterol; CHOL, cholesterol; TG, triglycerides; GLU, glucose; ALB, albumin; PRO, protein; Δ (CI), confidence intervals change; *p**, significant difference; rANOVA, repeated measures analysis of variance.

## Data Availability

All datasets presented in this study are included in the article/supplementary material.

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
