# Peer review of "Bariatric Surgery Induced Changes in Blood Cholesterol Are Modulated by Vitamin D Status"

_nutrients, 2022, doi:10.3390/nu14102000_

Round 1

Reviewer 1 Report

Bariatric surgery is a surgical procedure to help obese individuals to lose weight who are potentially at the risk of weight related health issues. Although it is one of the effective treatments for the obese individuals the effect of this procedure on the lipid metabolic parameters is not conclusive. Here Reczkowicz et al have proposed a beneficial role for Vitamin D levels post bariatric surgery. Authors found that post

 laparoscopic one anastomosis gastric bypass (OAGB) operation TG, hsCRP levels were decreased and HDL levels were elevated. Based on these results authors propose a protective role for vitamin D post OAGB surgical procedure.

This is a well-designed study. Authors have measured vitamin D levels and lipid metabolic parameters before, during and post-surgery were collected. One of the strong points of the paper is OAGB procedure did not have an effect on Vitamin D levels.

Major concerns:

  1. The ratio of male to female in the study was very less. Therefore sex specific claims cannot be made but it would have been nice to have equal number of males in the group.
  2. Method section is very descriptive. Briefly describe how serum was collected and lipid profile, Vitamin D, hsCRP, glucose, albumin and protein levels were measured?

Minor concerns:

  1. In Page 4 line 161-164 were merged with table. Align the table appropriately.
  2. You might want to have the title that directly describes your findings eg: “Vitamin D levels have a positive effects on lipids profile post bariatric surgery in humans”

Author Response

  1. The ratio of male to female in the study was very less. Therefore sex specific claims cannot be made but it would have been nice to have equal number of males in the group.

Answer:

It is currently impossible, due to the non-standard study of vitamin D metabolites, to complete the number of participants in the study so that there are the same number of them from each sex. However, in order to establish a correlation between the concentration of vitamin d and its effect on the biochemical parameters occurring after bariatric surgery, an equal sex ratio is not relevant. The biochemical changes that have been of interest to the researchers are not significantly dependent on gender, because even with this unequal ratio, such characteristics have not been noticed.

  1. Method section is very descriptive. Briefly describe how serum was collected and lipid profile, Vitamin D, hsCRP, glucose, albumin and protein levels were measured?

Answer

The biochemical blood test was carried out by a certified clinical laboratory at the University Clinical Center in Gdansk as part of an internal order. For this purpose, venous arterial blood was taken and this material was then subjected to biochemical tests in the field of hsCRP, glucose and others. On the other hand, the study of vitamin D metabolites was carried out by the laboratory of Masdiag sp. z o.o. in Warsaw by Dr. Konrad Kowalski from biological material taken as capillary blood secured on tissue paper intended exclusively for this purpose. Some changes in the methods section have been introduced.

You might want to have the title that directly describes your findings eg: “Vitamin D levels have a positive effects on lipids profile post bariatric surgery in humans”

In our opinion both title  could be  good  

Reviewer 2 Report

The study analyzed the concentation of VitD, lipid profile and other metabolic parameters in 24 obese patients after OAGB at baseline 3 months before the surgery, at the time of surgery and 6 month later. It's interesting and may help us to understand the changes of metablic parameters by bariatric surgery. Here are some suggestions:

1. In abstract, the author say "...demonstrate the correlation of the high level of vitamin D and the postoperative lipid profile and other metabolic parameters". However, the study did not perform correlation statistical analysis, only did one way ANOVA. The expression is inaccurate.

2. The basic information of the 24 patients, as body weight, BMi, gender, may these factors affect the result of the present study. Please give the details.

3. Season may be a factor which influence the level of VitD, if possiple, please give the information in which month the vitD  was measured.

4. In the study, all of the patients received 75ug VitD after sugury, will this affect the analysis of the results?

5. There are many study discussed the relationship between VitD and lipid profile, including invesgated the effectiveness of vitamin D supplementation on lipid profile in patients with metabolic disease. It seems that the present stuy ingnore this. 

Author Response

  1. In abstract, the author say "...demonstrate the correlation of the high level of vitamin D and the postoperative lipid profile and other metabolic parameters". However, the study did not perform correlation statistical analysis, only did one way ANOVA. The expression is inaccurate.

Answer:

Thank you for the comment. We admit that the expression was inaccurate. We just wanted to highlight that changes in some metabolic parameters are related to baseline vitamin 25OHD3 level (Table 3).  We have changed the sentence

  1. The basic information of the 24 patients, as body weight, BMi, gender, may these factors affect the result of the present study. Please give the details.

Answer:

We have added somatic characteristics of participants. Additionally, we have added the analyses of the influence of weight and BMI on the results.

  1. Season may be a factor which influence the level of VitD, if possible, please give the information in which month the vitD  was measured.

Answer:

As a rule, only the appropriate UV wavelength associated with the seasons, which is variable in the geographical width of the Surgical Clinic of the University Clinical Center in Gdansk, where bariatric operations were performed, can affect the synthesis of vitamin D metabolites in the body. It should be borne in mind that this is only a generalization, because it is not the time of year itself that causes such an effect, it is also dependent on the type of skin, pigmentation or age. It should be noted that UV radiation associated with changes in seasons had no effect on the assumptions of this study. Assumption of the study showed that the level of vitamin D concentration above 32 ng / ml among bariatric patients causes the expected biochemical postoperative effect. Therefore, it does not matter whether it was caused by supplementation or maybe natural factors related to the season.

  1. In the study, all of the patients received 75ug VitD after surgery, will this affect the analysis of the results?

Answer:

In this study, the concentration of vitamin D before surgery and the level of its impact on metabolic factors after bariatric surgery was analyzed. Bariatric patients both before and after surgery are subject to standard procedures, also in the field of supplementation. The supplement used, and thus the daily dose of vit. D after surgery is the same for all patients. In order to be able to determine whether postoperative supplementation at a dose of 75 ug vit D per day could cause the effect indicated in the study, the amount of this dose should be individually adjusted to a specific patient so as to achieve a minimum of 32 ng / ml of vitamin D metabolites in his body. The procedure of bariatric surgery does not include such parameters. It is therefore impossible to determine whether this dose could have achieved the postoperative metabolic effect indicated in the study.  

  1. There are many study discussed the relationship between VitD and lipid profile, including investigated the effectiveness of vitamin D supplementation on lipid profile in patients with metabolic disease. It seems that the present study ignore this.

Answer:

Indeed, it is expedient to add these issues. The part of the publication intended for discussion has been supplemented with appropriate references. We hope that this would be a sufficient complement. 

Round 2

Reviewer 2 Report

  1. Please show the results of covariance (ANCOVA) analyse to adjust between-measurements effects by weight and BMI as supplemental data.
  2. It's interesting to see whether there is a relationship between baseline VitD and metabolic paremeters, including body weight and BMI. Please add these analyse data.

Author Response

R: Please show the results of covariance (ANCOVA) analyse to adjust between-measurements effects by weight and BMI as supplemental data. A: We have added the Table presenting the suggested analysis as a supplement. Supplement Table 1. Intervention results for biochemistry parameters adjusted to baseline values of BMI and body weight. Covariant BMI Body weight ANCOVA p η_p^2 p η_p^2 hsCRP mg/L] 0.01 0.18 0.01 0.23 LDL [mg/dL] 0.81 0.01 0.80 0.01 HDLC[mg/dL] 0.10 0.07 0.04 0.10 CHOL[mg/dL] 0.10 0.07 0.07 0.08 TG [mg/dL] 0.10 0.07 0.06 0.09 GLU [mg/dL] 0.00 0.26 0.00 0.27 ALB [g/L] 0.01 0.17 0.00 0.25 PRO [g/L] 0.09 0.08 0.02 0.13 Note: hsCRP, C-reactive protein highly sensitive; LDLC, low-density lipoprotein cholesterol; HDLC, high-density lipoprotein cholesterol; CHOL, cholesterol; TG, triglycerides; GLU, glucose; ALB, albumin; PRO, protein; ANCOVA, analysis of covariance. R: It's interesting to see whether there is a relationship between baseline VitD and metabolic paremeters, including body weight and BMI. Please add these analyse data. A: We have added the correlation as a supplemental Table. Supplement Table 2. Correlation between baseline vitamin D and metabolic parameters. Baseline Weight (B) Weight (C) BMI (B) BMI (C) 25(OH)D3 [ng/mL] -0,07 -0,12 -0,05 0,11 25(OH)D2 [ng/mL] -0,10 -0,17 -0,12 -0,23 24,25(OH)2D3 [ng/mL] -0,04 -0,10 -0,08 0,18 epi-25(OH)D3 [ng/mL] -0,18 -0,13 -0,10 0,06